# DEEP COHERENT EXPLORATION FOR CONTINUOUS CONTROL

## ABSTRACT

In policy search methods for reinforcement learning (RL), exploration is often performed by injecting noise either in action space at each step independently or in parameter space over each full trajectory. In prior work, it has been shown that with linear policies, a more balanced trade-off between these two exploration strategies is beneficial. However, that method did not scale to policies using deep neural networks. In this paper, we introduce Deep Coherent Exploration, a general and scalable exploration framework for deep RL algorithms on continuous control, that generalizes step-based and trajectory-based exploration. This framework models the last layer parameters of the policy network as latent variables and uses a recursive inference step within the policy update to handle these latent variables in a scalable manner. We find that Deep Coherent Exploration improves the speed and stability of learning of A2C, PPO, and SAC on several continuous control tasks.

## 1 INTRODUCTION

The balance of exploration and exploitation (Kearns & Singh, 2002; Jaksch et al., 2010) is a long-standing challenge in reinforcement learning (RL). With insufficient exploration, states and actions with high rewards can be missed, resulting in policies prematurely converging to bad local optima. In contrast, with too much exploration, agents could waste their resources trying suboptimal states and actions, without leveraging their experiences efficiently. To learn successful strategies, this trade-off between exploration and exploitation must be balanced well, and this is known as the *exploration vs. exploitation dilemma*.

At a high level, exploration can be divided into directed strategies and undirected strategies (Thrun, 1992; Plappert et al., 2018). While directed strategies aim to extract useful information from existing experiences for better exploration, undirected strategies rely on injecting randomness into the agent's decision-making. Over the years, many sophisticated directed exploration strategies have been proposed (Tang et al., 2016; Ostrovski et al., 2017; Houthooft et al., 2016; Pathak et al., 2017). However, since these strategies still require lower-level exploration to collect the experiences, or are either complicated or computationally intensive, undirected exploration strategies are still commonly used in RL literature in practice, where some well-known examples are $\epsilon$-greedy (Sutton, 1995) for discrete action space and additive Gaussian noise for continuous action space (Williams, 1992). Such strategies explore by randomly perturbing agents' actions at different steps independently and hence are referred to as performing *step-based* exploration in action space (Deisenroth et al., 2013).

As an alternative to those exploration strategies in action space, exploration by perturbing the weights of linear policies has been proposed (Rückstieß et al., 2010; Sehnke et al., 2010; Kober & Peters, 2008). Since these strategies in parameter space naturally explore conditioned on the states and are usually *trajectory-based* (only perturb the weights at the beginning of each trajectory) (Deisenroth et al., 2013), they have the advantages of being more consistent, structured, and global (Deisenroth et al., 2013). Later, van Hoof et al. (2017) proposed a generalized exploration (GE) scheme, bridging the gap between step-based and trajectory-based exploration in parameter space. With the advance of deep RL, NoisyNet (Fortunato et al., 2018) and Parameter Space Noise for Exploration (PSNE) (Plappert et al., 2018) were introduced, extending parameter-space exploration strategies for policies using deep neural networks.

Although GE, NoisyNet, and PSNE improved over the vanilla exploration strategies in parameter space and were shown leading to more global and consistent exploration, they still suffer from several limitations. Given this, we propose a new exploration scheme with the following characteristics.

1. **Generalizing Step-based and Trajectory-based Exploration (van Hoof et al., 2017)** Since both NoisyNet and PSNE are trajectory-based exploration strategies, they are considered relatively inefficient and bring insufficient stochasticity (Deisenroth et al., 2013). Following van Hoof et al. (2017), our method improves by interpolating between step-based and trajectory-based exploration in parameter space, where a more balanced trade-off between stability and stochasticity can be achieved.

2. **Recursive Analytical Integration of Latent Exploring Policies** NoisyNet and PSNE address the uncertainty from sampling exploring policies using Monte Carlo integration, while GE uses analytical integration on full trajectories, which scales poorly in the number of time steps. In contrast, we apply analytical and recurrent integration after each step, which leads to low-variance and scalable updates.

3. **Perturbing Last Layers of Policy Networks** Both NoisyNet and PSNE perturb all layers of the policy network. However, in general, only the uncertainty in parameters of the last (linear) layer can be integrated analytically. Furthermore, it is not clear that deep neural networks can be perturbed in meaningful ways for exploration (Plappert et al., 2018). We thus propose and evaluate an architecture where perturbation is only applied on the parameters of the last layer.

These characteristics define our contribution, which we will refer to as Deep Coherent Exploration. We evaluate the coherent versions of A2C (Mnih et al., 2016), PPO (Schulman et al., 2017), and SAC (Haarnoja et al., 2018), where the experiments on OpenAI MuJoCo (Todorov et al., 2012; Brockman et al., 2016) tasks show that Deep Coherent Exploration outperforms other exploration strategies in terms of both learning speed and stability.

## 2 RELATED WORK

As discussed, exploration can broadly be classified into directed and undirected strategies (Thrun, 1992; Plappert et al., 2018), with undirected strategies being commonly used in practice because of their simplicity. Well known methods such as $\epsilon$-greedy (Sutton, 1995) or additive Gaussian noise (Williams, 1992) randomly perturb the action at each time step independently. These high-frequency perturbations, however, can result in poor coverage of the state-action space due to random-walk behavior (Rückstieß et al., 2010; Deisenroth et al., 2013), washing-out of exploration by the environment dynamics (Kober & Peters, 2008; Rückstieß et al., 2010; Deisenroth et al., 2013), and to potential damage to mechanical systems (Koryakovskiy et al., 2017).

One alternative is to instead perturb the policy in parameter space, with the perturbation held constant for the duration of a trajectory. Rückstieß et al. (2010) and Sehnke et al. (2010) showed that such parameter-space methods could bring improved exploration behaviors because of reduced variance and faster convergence, when combined with REINFORCE (Williams, 1992) or Natural Actor-Critic (Peters et al., 2005).

Another alternative to independent action-space perturbation, is to correlate the noise applied at subsequent actions (Morimoto & Doya, 2000; Wawrzynski, 2015; Lillicrap et al., 2016), for example by generating perturbations from an Ornstein-Uhlenbeck (OU) process (Uhlenbeck & Ornstein, 1930). Later, van Hoof et al. (2017) used the same stochastic process but in the parameter space of the policy. This approach uses a temporally coherent exploring policy, which unifies step-based and trajectory-based exploration. Moreover, the author showed that, with linear policies, a more delicate balance between these two extreme strategies could have better performance. However, this approach was derived in a batch mode setting and requires storing the full trajectory history and the inversion of a matrix growing with the number of time step. Thus, it does not scale well to long trajectories or complex models.

Although these methods pioneered the research of exploration in parameter space, their applicability is limited. More precisely, these methods were only evaluated with extremely shallow (often linear) policies and relatively simple tasks with low-dimensional state spaces and action spaces. Given this,

NoisyNet (Fortunato et al., 2018), PSNE (Plappert et al., 2018) and Stochastic A3C (SA3C) (Shang et al., 2019) were proposed, introducing more general and scalable methods for deep RL algorithms.

All three of these methods can be seen as learning a distribution over policies for trajectory-based exploration in parameter space. These exploring policies are sampled by perturbing the weights across all layers of a deep neural network, with the uncertainty from sampling being addressed by Monte Carlo integration. Whereas NoisyNet learns the magnitudes of the noise for each parameter, PSNE heuristically adapts a single magnitude for all parameters.

While showing good performance in practice (Fortunato et al., 2018; Plappert et al., 2018), these methods suffer from two potential limitations. Firstly, trajectory-based strategies can be inefficient as only one strategy can be evaluated for a potentially long trajectory (Deisenroth et al., 2013), which could result in a failure to escape local optima. Secondly, Monte Carlo integration results in high-variance gradient estimates which could lead to oscillating updates.

## 3 BACKGROUND

This section provides background for reinforcement learning and related deep RL algorithms.

### 3.1 REINFORCEMENT LEARNING

Reinforcement learning is a sub-field of machine learning that studies how an agent learns strategies with high returns through trial-and-error by interacting with an environment. This interaction between an agent and an environment is described using Markov Decision Processes (MDPs). A MDP is a tuple $(\mathcal{S}, \mathcal{A}, r, P, \gamma)$, where $\mathcal{S}$ is the state space, $\mathcal{A}$ is the action space, $r : \mathcal{S} \times \mathcal{A} \times \mathcal{S} \to \mathbb{R}$ is the reward function with $r_t = r(\mathbf{s}_t, \mathbf{a}_t, \mathbf{s}_{t+1})$, $P : \mathcal{S} \times \mathcal{A} \times \mathcal{S} \to \mathbb{R}^+$ is the transition probability function, and $\gamma$ is a discount factor indicating the preference of short-term rewards.

In RL with continuous action space, an agent aims to learn a parametrized (e.g. Gaussian) policy $\pi_{\boldsymbol{\theta}}(\mathbf{a}|\mathbf{s}) : \mathcal{S} \times \mathcal{A} \to \mathbb{R}^+$, with parameters $\boldsymbol{\theta}$, that maximizes the expected return over trajectories:

$$J(\boldsymbol{\theta}) = \mathbb{E}_{\tau \sim p(\tau|\pi_{\boldsymbol{\theta}})}[R(\tau)], \tag{1}$$

where $\tau = (\mathbf{s}_0, \mathbf{a}_0, ..., \mathbf{a}_{T-1}, \mathbf{s}_T)$ is a trajectory and $R(\tau) = \sum_{t=0}^{T} \gamma^t r_t$ is the discounted return.

### 3.2 DEEP REINFORCEMENT LEANING ALGORITHMS

Deep reinforcement learning combines deep learning and reinforcement learning, where policies and value functions are represented by deep neural networks for more sophisticated and powerful function approximation. In our experiments, we consider the following three deep RL algorithms.

**Advantage Actor-Critic (A2C)** Closely related to REINFORCE (Williams, 1992), A2C is an on-policy algorithm proposed as the synchronous version of the original Asynchronous Advantage Actor-Critic (A3C) (Mnih et al., 2016). The gradient of A2C can be written as:

$$\nabla_{\boldsymbol{\theta}} J(\boldsymbol{\theta}) = \mathbb{E}_{\tau \sim p(\tau|\boldsymbol{\theta})} \left[ \sum_{t=0}^{T-1} \nabla_{\boldsymbol{\theta}} \log \pi_{\boldsymbol{\theta}}(\mathbf{a}_t|\mathbf{s}_t) A^{\pi_{\boldsymbol{\theta}}}(\mathbf{s}_t, \mathbf{a}_t) \right], \tag{2}$$

where $A^{\pi_{\boldsymbol{\theta}}}(\mathbf{s}_t, \mathbf{a}_t)$ is the estimated advantage following policy $\pi_{\boldsymbol{\theta}}$.

**Proximal Policy Optimization (PPO)** PPO is an on-policy algorithm developed to determine the largest step for update while still keeping the updated policy close to the old policy in terms of Kullback–Leibler (KL) divergence. Instead of using a second-order method as in Trust Region Policy Optimization (TRPO) (Schulman et al., 2015), PPO applies a first-order method and combines several tricks to relieve the complexity. We consider the primary variant of PPO-Clip with the following surrogate objective:

$$L_{\boldsymbol{\theta}_k}^{CLIP}(\boldsymbol{\theta}) = \mathbb{E}_{\tau \sim p(\tau|\boldsymbol{\theta}_k)} \left[ \sum_{t=0}^{T-1} \left[ \min\left(r_t(\boldsymbol{\theta}), \text{clip}\left(r_t(\boldsymbol{\theta}), 1 - \epsilon, 1 + \epsilon\right)\right) A_t^{\pi_{\boldsymbol{\theta}_k}} \right] \right], \tag{3}$$

where $r_t(\boldsymbol{\theta}) = \dfrac{\pi_{\boldsymbol{\theta}}\left(\mathbf{a}_t|\mathbf{s}_t\right)}{\pi_{\boldsymbol{\theta}_k}\left(\mathbf{a}_t|\mathbf{s}_t\right)}$ and $\epsilon$ is a small threshold that approximately restricts the distance between the new policy and the old policy. In practice, to prevent the new policy from changing too fast, the KL divergence from the new policy to the old policy approximated on a sampled batch is often used as a further constraint.

**Soft Actor-Critic (SAC)**    As an entropy-regularized (Ziebart et al., 2008) off-policy actor-critic method (Lillicrap et al., 2016; Fujimoto et al., 2018) with a stochastic policy, SAC (Haarnoja et al., 2018) learns the optimal entropy-regularized $Q$-function through 'soft' Bellman back-ups with off-policy data:

$$Q^{\pi}(\mathbf{s}, \mathbf{a}) = \mathbb{E}_{\mathbf{s}' \sim p(\mathbf{s}'|\mathbf{s},\mathbf{a}), \tilde{\mathbf{a}}' \sim \pi(\tilde{\mathbf{a}}'|\mathbf{s}')} \left[ r + \gamma \left( Q^{\pi}\left(\mathbf{s}', \tilde{\mathbf{a}}'\right) + \alpha H\left(\pi\left(\tilde{\mathbf{a}}'|\mathbf{s}'\right)\right)\right)\right], \tag{4}$$

where $H$ is the entropy and $\alpha$ is the temperature parameter. The policy is then learned by maximizing the expected maximum entropy $V$-function via the reparameterization trick (Kingma et al., 2015).

## 4    DEEP COHERENT EXPLORATION FOR CONTINUOUS CONTROL

To achieve the desiderata in Section 1, we propose Deep Coherent Exploration, a method that models the policy as a generative model with latent variables. This policy is represented as $\pi_{\mathbf{w}_t,\boldsymbol{\theta}}(\mathbf{a}_t|\mathbf{s}_t) = \mathcal{N}(\mathbf{W}_t f_{\boldsymbol{\theta}}(\mathbf{s}_t) + \mathbf{b}_t, \boldsymbol{\Lambda}_a^{-1})$. Here $\mathbf{w}_t$ denotes all last layer parameters of policy network at step $t$ by combining $\mathbf{W}_t$ and $\mathbf{b}_t$, $\boldsymbol{\theta}$ denotes the parameters of the policy network except for the last layer, and $\boldsymbol{\Lambda}_a$ is a fixed and diagonal precision matrix. Our method treats the last layer parameters $\mathbf{w}_t$ as latent variables with marginal distribution $\mathbf{w}_t \sim \mathcal{N}\left(\boldsymbol{\mu}_t, \boldsymbol{\Lambda}_t^{-1}\right)$, where $\boldsymbol{\mu}_t$ and $\boldsymbol{\Lambda}_t$ are functions of learnable parameters $\boldsymbol{\mu}$ and $\boldsymbol{\Lambda}$ respectively. In this model, all learnable parameters can be denoted as $\zeta = (\boldsymbol{\mu}, \boldsymbol{\Lambda}, \boldsymbol{\theta})$. We provide a graphical model of Deep Coherent Exploration in Appendix A.

As in van Hoof et al. (2017), Deep Coherent Exploration generalizes step-based and trajectory-based exploration by constructing a Markov chain of $\mathbf{w}_t$. This Markov chain specifies joint probabilities through an initial distribution $p_0(\mathbf{w}_0) = \mathcal{N}\left(\boldsymbol{\mu}, \boldsymbol{\Lambda}^{-1}\right)$ and the conditional distribution $p(\mathbf{w}_t|\mathbf{w}_{t-1})$. This latter term explicitly expresses temporal coherence between subsequent parameter vectors. In this setting, step-based exploration corresponds to the extreme case when $p(\mathbf{w}_t|\mathbf{w}_{t-1}) = p_0(\mathbf{w}_t)$, and trajectory-based exploration corresponds to another extreme case when $p(\mathbf{w}_t|\mathbf{w}_{t-1}) = \delta(\mathbf{w}_t - \mathbf{w}_{t-1})$, where $\delta$ is the Dirac delta function. To ensure the marginal distribution of $\mathbf{w}_t$ will be equal to the initial distribution $p_0$ at any step $t$, we directly follow van Hoof et al. (2017) with the following transition distribution for $\mathbf{w}_t$:

$$p(\mathbf{w}_t|\mathbf{w}_{t-1}) = \mathcal{N}\left((1-\beta)\,\mathbf{w}_{t-1} + \beta\boldsymbol{\mu}, (2\beta - \beta^2)\boldsymbol{\Lambda}^{-1}\right), \tag{5}$$

where $\beta$ is a hyperparameter that controls the temporal coherency of $\mathbf{w}_t$ and $\mathbf{w}_{t-1}$. Then, the two extreme cases corresponds to $\beta = 0$ for trajectory-based exploration and $\beta = 1$ for step-based exploration, while the intermediate exploration corresponds to $\beta \in (0, 1)$. For intermediate values of $\beta$, we obtain smoothly changing policies that sufficiently explore, while reducing high-frequency perturbations.

### 4.1    ON-POLICY DEEP COHERENT EXPLORATION

Our method can be combined with all on-policy policy gradient methods and here we present this adaptation with REINFORCE (Williams, 1992). Starting from the RL objective in Equation 1:

$$\nabla_{\zeta} J(\zeta) = \mathbb{E}_{\tau \sim p(\tau|\zeta)} \left[\nabla_{\zeta} \log p(\tau|\zeta) R(\tau)\right], \tag{6}$$

the gradients w.r.t the sampled trajectory can be obtained using standard chain rule:

$$\nabla_{\zeta} \log p(\tau|\zeta) = \sum_{t=0}^{T-1} \left(\nabla_{\zeta} \log p(\mathbf{a}_t|\mathbf{s}_{[0:t]}, \mathbf{a}_{[0:t-1]}, \zeta)\right). \tag{7}$$

Here, since information can still flow through the unobserved latent variable $\mathbf{w}_t$, our policy is *not* Markov anymore. To simplify this dependency, we introduce $\mathbf{w}_t$ into $p(\mathbf{a}_t|\mathbf{s}_{[0:t]}, \mathbf{a}_{[0:t-1]}, \zeta)$:

$$p(\mathbf{a}_t|\mathbf{s}_{[0:t]}, \mathbf{a}_{[0:t-1]}, \zeta) = \int p(\mathbf{a}_t, \mathbf{w}_t|\mathbf{s}_{[0:t]}, \mathbf{a}_{[0:t-1]}, \zeta)d\mathbf{w}_t \tag{8}$$

$$= \int \underbrace{p(\mathbf{a}_t|\mathbf{s}_t; \mathbf{w}_t, \boldsymbol{\theta})}_{\text{Gaussian policy } \pi_{\mathbf{w}_t, \boldsymbol{\theta}}(\mathbf{a}_t|\mathbf{s}_t)} \underbrace{p(\mathbf{w}_t|\mathbf{s}_{[0:t-1]}, \mathbf{a}_{[0:t-1]}, \boldsymbol{\mu}, \boldsymbol{\Lambda}, \boldsymbol{\theta})}_{\text{forward message } \alpha(\mathbf{w}_t)} d\mathbf{w}_t, \tag{9}$$

where the first factor is the Gaussian action probability and the second factor can be interpreted as the forward message $\alpha(\mathbf{w}_t)$ along the chain. We decompose $\alpha(\mathbf{w}_t)$ by introducing $\mathbf{w}_{t-1}$ :

$$\alpha(\mathbf{w}_t) = \int p(\mathbf{w}_t, \mathbf{w}_{t-1}|\mathbf{s}_{[0:t-1]}, \mathbf{a}_{[0:t-1]}, \zeta)d\mathbf{w}_{t-1} \tag{10}$$

$$= \int \underbrace{p(\mathbf{w}_t|\mathbf{w}_{t-1}; \boldsymbol{\mu}, \boldsymbol{\Lambda})}_{\text{transition probability of } \mathbf{w}_t} \frac{p(\mathbf{a}_{t-1}|\mathbf{s}_{t-1}; \mathbf{w}_{t-1}, \boldsymbol{\theta})\alpha(\mathbf{w}_{t-1})}{\mathbf{Z}_{t-1}}d\mathbf{w}_{t-1}, \tag{11}$$

where the first factor is the transition probability of $\mathbf{w}_t$ (Equation 5) and $\mathbf{Z}_{t-1}$ in the factor term is a normalizing constant. Since Gaussians are closed under marginalization and condition, the second factor can be obtained analytically without the need of computing the normalizing constant $\mathbf{Z}_{t-1}$. Moreover, $\alpha(\mathbf{w}_{t-1})$ is a Gaussian by mathematical induction from the initial step. As a result, we arrive at an efficient recursive expression for exact inference of $\mathbf{w}_t$. Again, with the property of Gaussians, all integrals appearing above can be solved analytically, where the marginal action probability given the history at each step $t$ can be obtained and used for policy updates.

Summarizing, non-Markov policies require substituting the regular $p(\mathbf{a}_t|\mathbf{s}_t; \boldsymbol{\theta})$ term in the update equation with $p(\mathbf{a}_t|\mathbf{s}_{[0:t]}, \mathbf{a}_{[0:t-1]}, \zeta)$ (Equation 7). Equations 8-11 show how this expression can be efficiently calculated recursively. Except for this substitution, learning algorithms like A2C and PPO can proceed as normal. For detailed mathematical derivation, please refer to Appendix B.

## 4.2 OFF-POLICY DEEP COHERENT EXPLORATION

Combining our method with off-policy methods (Lillicrap et al., 2016; Fujimoto et al., 2018; Haarnoja et al., 2018) requires defining both the behavior policy and the update equation. The behavior policy is the same as in the on-policy methods discussed earlier (Equation 5). The policy update procedure may require adjustments for specific algorithms. Here, we show how to adapt our method for SAC (Haarnoja et al., 2018). In the SAC policy update, the target policy is adapted towards the exponential of the new $Q$-function. The target policy here is the marginal policy, rather than the policy conditioned on the sampled $\mathbf{w}$, as this second option would ignore the dependence on $\boldsymbol{\mu}$ and $\boldsymbol{\Lambda}$. This consideration leads to the following objective for policy update:

$$J(\zeta) = \mathbb{E}_{\mathbf{s}_t \sim \mathcal{D}} \left[ \text{KL} \left( p(\mathbf{a}_t|\mathbf{s}_t, \zeta) \,\Big\|\, \frac{\exp(Q_{\phi}(\mathbf{s}_t, \mathbf{a}_t))}{Z_{\phi}(\mathbf{s}_t)} \right) \right] \tag{12}$$

$$= \mathbb{E}_{\mathbf{s}_t \sim \mathcal{D}, \mathbf{a}_t \sim p(\mathbf{a}_t|\mathbf{s}_t, \zeta)} \left[ \log p(\mathbf{a}_t|\mathbf{s}_t, \zeta) - Q_{\phi}(\mathbf{s}_t, \mathbf{a}_t) \right], \tag{13}$$

where $\phi$ denotes the parameters of the $Q$-function and $p(\mathbf{a}_t|\mathbf{s}_t, \zeta)$ is the marginal policy which can again be obtained analytically (Equation 22):

$$p(\mathbf{a}_t|\mathbf{s}_t, \zeta) = \int \underbrace{p(\mathbf{a}_t|\mathbf{s}_t; \mathbf{w}_0, \boldsymbol{\theta})}_{\text{Gaussian policy } \pi_{\mathbf{w}_0, \boldsymbol{\theta}}(\mathbf{a}_t|\mathbf{s}_t)} \underbrace{p(\mathbf{w}_0|\boldsymbol{\mu}, \boldsymbol{\Lambda})}_{\text{marginal probability of } \mathbf{w}_0} d\mathbf{w}_0, \tag{14}$$

where all parameters can be learned via the reparameterization trick (Kingma et al., 2015).

## 5 EXPERIMENTS

For the experiments, we compare our method with NoisyNet (Fortunato et al., 2018), PSNE (Plappert et al., 2018) and standard action noise. This comparison is evaluated in combination of A2C (Mnih et al., 2016), PPO (Schulman et al., 2017) and SAC (Haarnoja et al., 2018) on OpenAI Gym MuJoCo (Todorov et al., 2012; Brockman et al., 2016) continuous control tasks.

For exploration in parameter space, we use a fixed action noise with a standard deviation of 0.1. For A2C and PPO, their standard deviations of parameter noise are all initialized at 0.017, as suggested in Fortunato et al. (2018). For SAC, we initialize the standard deviation of parameter noise at 0.034 for both our method and PSNE as it gave better results in practice. Besides, Deep Coherent Exploration learns the logarithm of parameter noise, while NoisyNet learns the parameter noise directly, and PSNE adapts the parameter noise. To protect the policies from changing too dramatically, we consider three small values of $\beta$ (0.0, 0.01, and 0.1) for Deep Coherent Exploration, where we use $\beta = 0.01$ for comparative evaluation with other exploration strategies. Our implementation of NoisyNet is based on the code from Kaixhin[1]. For PSNE, we refer to author's implementation in OpenAI Baselines[2] (Dhariwal et al., 2017) and the original paper, where we set the KL threshold for A2C and PPO to 0.01 and the MSE threshold for SAC to 0.1. On the other hand, exploration with action noise uses the default setting proposed by Achiam (2018), where the standard deviation of action noise is initialized at around 0.6 for A2C and PPO. For SAC, in the baseline setting the standard deviation of action noise is output by the policy network.

In all experiments, agents are trained with a total of $10^6$ environmental steps, where they are updated after each epoch. A2C and PPO use four parallel workers, where each worker collects a trajectory of 1000 steps for each epoch, resulting in epochs with 4000 steps in total. After each epoch, both A2C and PPO update their value functions for 80 gradient steps. At the same time, A2C updates its policy for one gradient step, while PPO updates its policy for up to 80 gradient steps until the KL constraint is satisfied. SAC uses a single worker, with a step size of 4000 for each epoch. After every 50 environmental steps, both policy and value functions are updated for 50 gradient steps. To make the parameter noise stable, we adapt the standard deviation of parameter noise after each epoch.

Our implementation of A2C, PPO, and SAC with different exploration strategies are adapted based on OpenAI Spinning Up (Achiam, 2018) with default settings. All three algorithms use two-layer feedforward neural networks with the same network architectures for both policy and value function. To be more precise, A2C and PPO use network architectures with 64 and 64 hidden nodes activated by tanh units. In comparison, SAC uses a network architecture of 256 and 256 hidden nodes activated by rectified linear units (ReLU). Parameters of policies and value functions in all three algorithms are updated using Adam (Kingma & Ba, 2015). A2C and PPO use a learning rate of $3 \cdot 10^{-4}$ for the policies and a learning rate of $10^{-3}$ for the value functions. SAC uses a single learning rate of $10^{-3}$ for both policy and value function.

For each task, we evaluate the performance of agents after every 5 epochs, with no exploration noise. Additionally, each evaluation reports the average reward of 10 episodes for each worker. To mitigate the randomness within environments and policies, we report our results as the average over 10 random seeds. All settings not explicitly explained in this section, are set to the code base defaults. For more details, please refer to documents and source code from OpenAI Spinning Up (Achiam, 2018) and our implementation[3].

## 5.1 COMPARATIVE EVALUATION

In this section, we present the results for A2C (Mnih et al., 2016), PPO (Schulman et al., 2017), and SAC (Haarnoja et al., 2018) on three control tasks, with the additional results shown in Appendix D. Figure 1 shows that, overall, Coherent-A2C outperforms all other A2C-based methods in terms of learning speed, final performance, and algorithm stability. In particular, given that Ant-v2 is considered a challenging task, Deep Coherent Exploration considerably accelerates learning speed. For PPO-based method, our method still outperforms NoisyNet and PSNE significantly in all tasks. However, our method's advantage compared to standard action noise is smaller. Particularly, Coherent-PPO underperforms PPO in Walker2d-v2. Two reasons might explain this. Firstly, some environments might be more unstable and require a larger degree of exploration, which favors PPO as it initializes its action noise with much greater value. Secondly, because of having extra parameters, policies of Coherent-PPO, NoisyNet-PPO, and PSNE-PPO tend to satisfy the KL constraint in fewer update steps, which leads to slower learning. For SAC, the advantages of our method are smaller compared to A2C and PPO. More specifically, Coherent-SAC learns slightly

---

[1] https://github.com/Kaixhin/NoisyNet-A3C
[2] https://github.com/openai/baselines/tree/master/baselines/ddpg
[3] Source code to be released after review

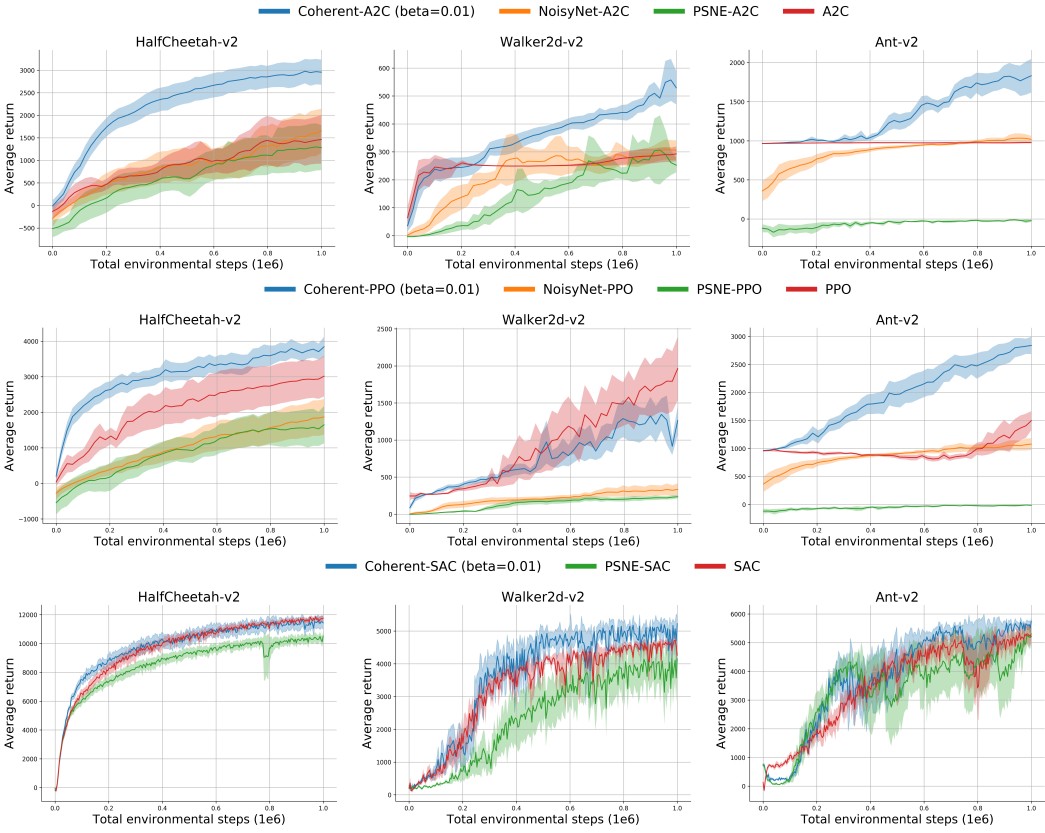

Figure 1: Learning curves for deep RL algorithms with different exploration strategies on OpenAI MuJoCo continuous control tasks, where the top, middle and bottom row corresponds to results of A2C, PPO, and SAC respectively. The solid curves correspond to the mean, and the shaped region represents half a standard deviation of the average return over 10 random seeds.

faster than SAC in HalfCheetah-v2 and achieves the highest average returns in Walker2d-v2 and Ant-v2. Furthermore, Coherent-SAC shows variance lower than PSNE-SAC but higher than the baseline SAC.

## 5.2 ABLATION STUDIES

In this section, we present three separate ablation studies to clarify the effect of each characteristic discussed in Section 1. These ablation studies are performed with A2C, to ensure that all characteristics are applicable and because the fixed number of gradient steps puts the methods on equal footing. Here we show the results for HalfCheetah-v2, where the full results can be found in Appendix D.

**Generalizing Step-based and Trajectory-based Exploration** As shown in Figure 2a, we evaluate three different values 0.0, 0.01, 0.1 of $\beta$ for Coherent-A2C. Here both two intermediate strategies ($\beta = 0.01$ and $\beta = 0.1$) outperforms the trajectory-based strategy ($\beta = 0.0$). Coherent-A2C with $\beta = 0.01$ seems to achieve the best balance between randomness and stability, with a considerably higher return than the other two.

**Analytical Integration of Latent Exploring Policies** We introduce OurNoisyNet for comparison. OurNoisyNet equips a noisy linear layer for only its last layer, and this layer learns the logarithm of standard deviation, as in Deep Coherent Exploration. We compare Coherent-A2C using $\beta = 0.0$ and OurNoisyNet-A2C, with the only difference thus being whether we integrate analytically or use the reparameterization trick (Kingma et al., 2015).

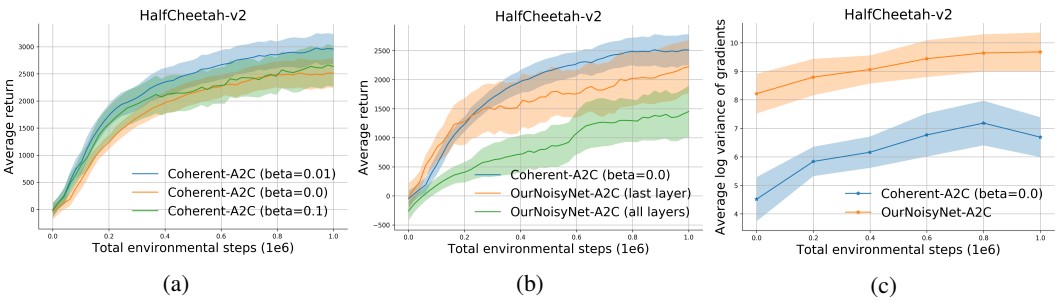

Figure 2: Results of Coherent-A2C with different settings for HalfCheetah-v2, where Figure 2a and Figure 2b show the learning curves and Figure 2c shows the average log variance of gradients during six stages in learning. The solid curves correspond to the mean, and the shaped region represents half a standard deviation of the average return over 10 random seeds.

We first measure the variance of gradient estimates in both methods. This variance is measured by computing the trace of the covariance matrix using 10 gradient samples. We report this measure in six stages during training, as shown in Figure 2c. We can observe that analytical integration leads to lower-variance gradient estimates across all training stages for HalfCheetah-v2. We further present the learning curves of both methods in Figure 2b, where Coherent-A2C with $\beta = 0.0$ shows higher return than OurNoisyNet-A2C. It is interesting that Coherent-A2C displays a much lower standard deviation across different random seeds. Furthermore, the lower-variance gradient estimates of Coherent-A2C could enable a larger learning rate for faster training without making policy updates unstable.

**Perturbing Last Layers of Policy Networks**   In this part, we compare OurNoisyNet-A2C perturbed over all layers, and OurNoisyNet-A2C perturbed over only the last layer. The result is shown in Figure 2b. Somewhat to our surprise, the latter seems to perform much better. There are several possible reasons. Firstly, since it is unknown how the parameter noise (especially in lower layers) is realized in action noise, perturbing all layers of the policy network may lead to uncontrollable perturbations. Such excess exploratory noise could inhibit exploitation. Secondly, perturbing all layers might disturb the representation learning of states, which is undesirable for learning a good policy. Thirdly, perturbing only the last layer could also lead to fewer parameters for NoisyNet.

## 6   CONCLUSION

In this paper, we have presented a general and scalable exploration framework that extends the generalized exploration scheme (van Hoof et al., 2017) for continuous deep RL algorithms. In particular, recursive calculation of marginal action probabilities allows handling long trajectories and high-dimensional parameter vectors. Compared with NoisyNet (Fortunato et al., 2018) and PSNE (Plappert et al., 2018), our method has three improvements. Firstly, Deep Coherent Exploration generalizes step-based and trajectory-based exploration in parameter space, which allows a more balanced trade-off between stochasticity and coherence. Secondly, Deep Coherent Exploration analytically marginalizes the latent policy parameters, yielding lower-variance gradient estimates that stabilize and accelerate learning. Thirdly, by perturbing only the last layer of the policy network, Deep Coherent Exploration provides better control of the injected noise.

When combining with A2C (Mnih et al., 2016), PPO (Schulman et al., 2017), and SAC (Haarnoja et al., 2018), we empirically show that Deep Coherent Exploration outperforms other exploration strategies on most of the MuJoCo continuous control tasks tested. Furthermore, the ablation studies show that, while each of the improvements is beneficial, combining them leads to even faster and more stable learning. For future work, since Deep Coherent Exploration uses a fixed and small action noise, we believe one interesting direction is to study whether the learnable perturbations in action space can be combined with our method in a meaningful way for even more effective exploration.

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

## A  GRAPHICAL MODEL OF DEEP COHERENT EXPLORATION

In this appendix, we provide a graphical model representation of Deep Coherent Exploration, shown in Figure 3. This graphical model uses the same conventions as in Bishop (2007), where empty circles denote latent random variables, shaded circles denote observed random variables, and dots denote deterministic variables.

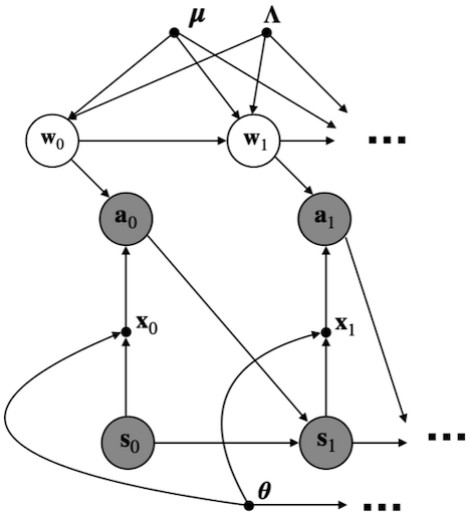

Figure 3: Graphical model of Deep Coherent Exploration.

## B  MARGINAL ACTION PROBABILITY FOR ON-POLICY COHERENT EXPLORATION

As discussed in Section 4.1, forward message $\alpha(\mathbf{w}_t)$ is used to compute the marginal action probability given the history at step $t$ for the final learning objective. Suppose we have the Gaussian policy as:

$$\pi_{\mathbf{w}_t, \boldsymbol{\theta}}(\mathbf{a}_t | \mathbf{s}_t) = \mathcal{N}(\mathbf{W}_t \mathbf{x}_t + \mathbf{b}_t, \boldsymbol{\Lambda}_a^{-1}), \tag{15}$$

where $\mathbf{a}_t \in \mathbb{R}^p$, $\mathbf{x}_t = f_{\boldsymbol{\theta}}(\mathbf{s}_t) \in \mathbb{R}^q$, $\mathbf{W}_t \in \mathbb{R}^{p \times q}$ is the coefficient matrix, $\mathbf{b}_t \in \mathbb{R}^p$ is the bias vector and $\boldsymbol{\Lambda}_a$ is a constant precision matrix for the Gaussian policy. It's helpful to represent $\mathbf{w}_t \in \mathbb{R}^{pq+p}$ by flattening $\mathbf{W}_t$ and combining $\mathbf{b}_t$:

$$\mathbf{w}_t = \begin{pmatrix} w_{11} \\ \vdots \\ w_{1q} \\ \vdots \\ w_{p1} \\ \vdots \\ w_{pq} \\ b_1 \\ \vdots \\ b_p \end{pmatrix}, \tag{16}$$

such that the parameters could still be sampled using multivariate Gaussians. Moreover, we stack $\mathbf{x}_t$ into $\mathbf{X}_t \in \mathbb{R}^{p \times (pq+p)}$:

$$\mathbf{X}_t = \begin{pmatrix} \mathbf{x}_t^T & \mathbf{0}_{q,1}^T & \cdots & \mathbf{0}_{q,1}^T & \mathbf{0}_{q,1}^T & 1 & 0 & \cdots & 0 & 0 \\ \mathbf{0}_{q,1}^T & \mathbf{x}_t^T & \cdots & \mathbf{0}_{q,1}^T & \mathbf{0}_{q,1}^T & 0 & 1 & \cdots & 0 & 0 \\ \vdots & \vdots & \ddots & \vdots & \vdots & \vdots & \vdots & \ddots & \vdots & \vdots \\ \mathbf{0}_{q,1}^T & \mathbf{0}_{q,1}^T & \cdots & \mathbf{x}_t^T & \mathbf{0}_{q,1}^T & 0 & 0 & \cdots & 1 & 0 \\ \mathbf{0}_{q,1}^T & \mathbf{0}_{q,1}^T & \cdots & \mathbf{0}_{q,1}^T & \mathbf{x}_t^T & 0 & 0 & \cdots & 0 & 1 \end{pmatrix}, \tag{17}$$

where $\mathbf{0}_{q,1}$ is a $q$-dimension zero column vector. After this transformation, the Gaussian policy is represented equivalently as:

$$\pi_{\mathbf{w}_t,\boldsymbol{\theta}}(\mathbf{a}_t|\mathbf{s}_t) = \mathcal{N}(\mathbf{X}_t\mathbf{w}_t, \boldsymbol{\Lambda}_a^{-1}). \tag{18}$$

### B.1 BASE CASE

For the base case $t = 0$, forward message $\alpha(\mathbf{w}_0)$ and the initial transition probability of $\mathbf{w}_0$ is identical by definition:

$$\alpha(\mathbf{w}_0) = p_0(\mathbf{w}_0; \boldsymbol{\mu}, \boldsymbol{\Lambda}) = \mathcal{N}\left(\boldsymbol{\mu}, \boldsymbol{\Lambda}^{-1}\right). \tag{19}$$

Additionally, the action probability is given by:

$$\pi_{\mathbf{w}_0,\boldsymbol{\theta}}(\mathbf{a}_0|\mathbf{s}_0) = \mathcal{N}(\mathbf{X}_0\mathbf{w}_0, \boldsymbol{\Lambda}_a^{-1}). \tag{20}$$

With the property of multivariate Gaussians, we obtain the marginal action probability given the history at $t = 0$:

$$\log p(\mathbf{a}_0|\mathbf{s}_0, \boldsymbol{\mu}, \boldsymbol{\Lambda}, \boldsymbol{\theta}) = \log \int \pi_{\mathbf{w}_0,\boldsymbol{\theta}}(\mathbf{a}_0|\mathbf{s}_0)\alpha(\mathbf{w}_0)d\mathbf{w}_0 \tag{21}$$

$$= \log \mathcal{N}(\mathbf{X}_0\boldsymbol{\mu}, \boldsymbol{\Lambda}_a^{-1} + \mathbf{X}_0\boldsymbol{\Lambda}^{-1}\mathbf{X}_0^T). \tag{22}$$

### B.2 GENERAL CASE

For the general case of step $t > 0$, we need the state $\mathbf{s}_{t-1}$, action $\mathbf{a}_{t-1}$ as well as mean and covariance of forward message $\alpha(\mathbf{w}_{t-1})$ stored from previous step. Suppose $\alpha(\mathbf{w}_{t-1})$ is written as:

$$\alpha(\mathbf{w}_{t-1}) = \mathcal{N}(\mathbf{v}_{t-1}, \mathbf{L}_{t-1}^{-1}), \tag{23}$$

and the action probability from the previous step is given by:

$$\pi_{\mathbf{w}_{t-1},\boldsymbol{\theta}}(\mathbf{a}_{t-1}|\mathbf{s}_{t-1}) = \mathcal{N}\left(\mathbf{X}_{t-1}\mathbf{w}_{t-1}, \boldsymbol{\Lambda}_a^{-1}\right). \tag{24}$$

We have directly:

$$p(\mathbf{w}_{t-1}|\mathbf{s}_{[0:t-1]}, \mathbf{a}_{[0:t-1]}, \boldsymbol{\mu}, \boldsymbol{\Lambda}, \boldsymbol{\theta}) = \mathcal{N}\left(\mathbf{u}_{t-1}, \boldsymbol{\Sigma}_{t-1}\right), \tag{25}$$

with

$$\mathbf{u}_{t-1} = \boldsymbol{\Sigma}_{t-1}\left(\mathbf{X}_{t-1}^T\boldsymbol{\Lambda}_a\mathbf{a}_{t-1} + \mathbf{L}_{t-1}\mathbf{v}_{t-1}\right) \tag{26}$$

$$\boldsymbol{\Sigma}_{t-1} = \left(\mathbf{L}_{t-1} + \mathbf{X}_{t-1}^T\boldsymbol{\Lambda}_a\mathbf{X}_{t-1}\right)^{-1}. \tag{27}$$

Combining the transition probability of $\mathbf{w}_t$:

$$p(\mathbf{w}_t|\mathbf{w}_{t-1}; \boldsymbol{\mu}, \boldsymbol{\Lambda}) = \mathcal{N}\left((1-\beta)\mathbf{w}_{t-1} + \beta\boldsymbol{\mu}, (2\beta - \beta^2)\boldsymbol{\Lambda}^{-1}\right), \tag{28}$$

we obtain the forward message $\alpha(\mathbf{w}_t)$:

$$\alpha\left(\mathbf{w}_t\right) = \mathcal{N}\left(\mathbf{v}_t, \mathbf{L}_t^{-1}\right), \tag{29}$$

where

$$\mathbf{v}_t = (1-\beta)\mathbf{u}_{t-1} + \beta\boldsymbol{\mu} \tag{30}$$

$$\mathbf{L}_t^{-1} = (2\beta - \beta^2)\boldsymbol{\Lambda}^{-1} + (1-\beta)^2\boldsymbol{\Sigma}_{t-1}. \tag{31}$$

Here, $\mathbf{v}_t$ and $\mathbf{L}_t^{-1}$ should be stored and used for exact inference of $\alpha(\mathbf{w}_{t+1})$ at the next step. Finally, the marginal action probability given the history at step $t > 0$ is given by:

$$\log p(\mathbf{a}_t|\mathbf{s}_{[0:t]}, \mathbf{a}_{[0:t-1]}, \boldsymbol{\mu}, \boldsymbol{\Lambda}, \boldsymbol{\theta}) = \log \int \pi_{\mathbf{w}_t,\boldsymbol{\theta}}(\mathbf{a}_t|\mathbf{s}_t)\alpha(\mathbf{w}_t)d\mathbf{w}_t \tag{32}$$

$$= \log \mathcal{N}(\mathbf{X}_t\mathbf{v}_t, \boldsymbol{\Lambda}_a^{-1} + \mathbf{X}_t\mathbf{L}_t^{-1}\mathbf{X}_t^T). \tag{33}$$

## C  DEEP COHERENT REINFORCEMENT LEARNING

Here, we provide a brief introduction of adapting Deep Coherent Exploration for A2C (Mnih et al., 2016), PPO (Schulman et al., 2017) and SAC (Haarnoja et al., 2018). Respectively, we call them Coherent-A2C, Coherent-PPO and Coherent-SAC.

### C.1  COHERENT ADVANTAGE ACTOR-CRITIC (COHERENT-A2C)

Coherent-A2C is straightforward to implement. To do that, one could just replace the original A2C gradient estimates $\hat{\nabla}_{\boldsymbol{\theta}} J(\boldsymbol{\theta})$ with the on-policy coherent gradient estimates $\hat{\nabla}_{\boldsymbol{\mu},\boldsymbol{\Lambda},\boldsymbol{\theta}} J(\boldsymbol{\mu},\boldsymbol{\Lambda},\boldsymbol{\theta})$. The pseudo-code of single-worker Coherent-A2C is shown in Algorithm 1.

---

**Algorithm 1:** Coherent-A2C

**Input:** initial policy parameters $\boldsymbol{\mu}_0, \boldsymbol{\Lambda}_0, \boldsymbol{\theta}_0$, initial value function parameters $\boldsymbol{\phi}_0$.

1 **for** $k$=0,1,2,...,$K$ **do**
2     Create a buffer $\mathcal{D}_k$ for collecting a trajectory $\tau_k$ with $T$ steps.
3     **for** $t$=0,...,$T$ **do**
4         **if** $t$=0 **then**
5             Sample last layer parameters of policy network $\mathbf{w}_t \sim p_0(\mathbf{w}_t; \boldsymbol{\mu}_k, \boldsymbol{\Lambda}_k)$ and store $\mathbf{w}_t$.
6         **else**
7             Sample last layer parameters of policy network $\mathbf{w}_t \sim p(\mathbf{w}_t | \mathbf{w}_{t-1}; \boldsymbol{\mu}_k, \boldsymbol{\Lambda}_k)$ and store $\mathbf{w}_t$.
8         Observe state $\mathbf{s}_t$ and select action $\mathbf{a}_t \sim \pi_{\mathbf{w}_t, \boldsymbol{\theta}_k}(\mathbf{a}_t | \mathbf{s}_t)$.
9         Execute $\mathbf{a}_t$ in the environment.
10         Observe next state $\mathbf{s}_{t+1}$, reward $r_t$, done signal $d$, and store $(\mathbf{s}_t, \mathbf{a}_t, r_t, \mathbf{s}_{t+1}, d)$ in buffer $\mathcal{D}_k$.
11         If $\mathbf{s}_{t+1}$ is terminal, reset environment state.
12         Infer forward message $\alpha(\mathbf{w}_t)$ using previous state $\mathbf{s}_{t-1}$, previous action $\mathbf{a}_{t-1}$ as well as mean $\mathbf{v}_{t-1}$ and covariance $\mathbf{L}_{t-1}^{-1}$ of previous forward message $\alpha(\mathbf{w}_{t-1})$.
13         Store mean $\mathbf{v}_t$ and covariance $\mathbf{L}_t^{-1}$ of current forward message $\alpha(\mathbf{w}_t)$.
14         Compute marginal action probability $p(\mathbf{a}_t | \mathbf{s}_{[0:t]}, \mathbf{a}_{[0:t-1]}, \boldsymbol{\mu}_k, \boldsymbol{\Lambda}_k, \boldsymbol{\theta}_k)$.
15     Compute rewards-to-go $R_t$ and any kind of advantage estimates $\hat{A}_t$ based on current value function $V_{\boldsymbol{\phi}_k}$ for all steps $t$.
16     Estimate gradient of the policy:

$$\hat{\nabla}_{\boldsymbol{\mu},\boldsymbol{\Lambda},\boldsymbol{\theta}} J(\boldsymbol{\mu},\boldsymbol{\Lambda},\boldsymbol{\theta}) = \sum_{t=0}^{T-1} \nabla_{\boldsymbol{\mu},\boldsymbol{\Lambda},\boldsymbol{\theta}} \log p(\mathbf{a}_t | \mathbf{s}_{[0:t]}, \mathbf{a}_{[0:t-1]}, \boldsymbol{\mu}_k, \boldsymbol{\Lambda}_k, \boldsymbol{\theta}_k) \hat{A}_t,$$

    and update the policy by performing a gradient step:

$$\boldsymbol{\mu}_{k+1} \leftarrow \boldsymbol{\mu}_k + \alpha_{\boldsymbol{\mu}} \hat{\nabla}_{\boldsymbol{\mu}} J(\boldsymbol{\mu},\boldsymbol{\Lambda},\boldsymbol{\theta})$$
$$\boldsymbol{\Lambda}_{k+1} \leftarrow \boldsymbol{\Lambda}_k + \alpha_{\boldsymbol{\Lambda}} \hat{\nabla}_{\boldsymbol{\Lambda}} J(\boldsymbol{\mu},\boldsymbol{\Lambda},\boldsymbol{\theta})$$
$$\boldsymbol{\theta}_{k+1} \leftarrow \boldsymbol{\theta}_k + \alpha_{\boldsymbol{\theta}} \hat{\nabla}_{\boldsymbol{\theta}} J(\boldsymbol{\mu},\boldsymbol{\Lambda},\boldsymbol{\theta}).$$

17     Learn value function by minimizing the regression mean-squared error:

$$L(\boldsymbol{\phi}) = \frac{1}{T} \sum_{t=0}^{T} \left( V_{\boldsymbol{\phi}_k}(\mathbf{s}_t) - R_t \right)^2,$$

    and update the value function by performing a gradient step:

$$\boldsymbol{\phi}_{k+1} \leftarrow \boldsymbol{\phi}_k + \alpha_{\boldsymbol{\phi}} \hat{\nabla}_{\boldsymbol{\phi}} L(\boldsymbol{\phi}).$$

---

## C.2 COHERENT PROXIMAL POLICY OPTIMIZATION (COHERENT-PPO)

Coherent-PPO can be implemented in a similar way. As in Coherent-A2C, we substitute the original objective $L_{\boldsymbol{\theta}_k}^{CLIP}(\boldsymbol{\theta})$ with $L_{\boldsymbol{\mu}_k, \boldsymbol{\Lambda}_k, \boldsymbol{\theta}_k}^{CLIP}(\boldsymbol{\mu}, \boldsymbol{\Lambda}, \boldsymbol{\theta})$, which is given by:

$$L_{\boldsymbol{\mu}_k, \boldsymbol{\Lambda}_k, \boldsymbol{\theta}_k}^{CLIP}(\boldsymbol{\mu}, \boldsymbol{\Lambda}, \boldsymbol{\theta}) \tag{34}$$

$$= \mathbb{E}_{\tau \sim p(\tau | \boldsymbol{\mu}_k, \boldsymbol{\Lambda}_k, \boldsymbol{\theta}_k)} \sum_{t=0}^{T-1} \left[ \min \left( r_t(\boldsymbol{\mu}, \boldsymbol{\Lambda}, \boldsymbol{\theta}), \text{clip} \left( r_t(\boldsymbol{\mu}, \boldsymbol{\Lambda}, \boldsymbol{\theta}), 1 - \epsilon, 1 + \epsilon \right) \right) A_t^{\pi_{\boldsymbol{\mu}, \boldsymbol{\Lambda}, \boldsymbol{\theta}}} \right], \tag{35}$$

where $r_t(\boldsymbol{\mu}, \boldsymbol{\Lambda}, \boldsymbol{\theta}) = \frac{p(\mathbf{a}_t | \mathbf{s}_{[0:t]}, \mathbf{a}_{[0:t-1]}, \boldsymbol{\mu}, \boldsymbol{\Lambda}, \boldsymbol{\theta})}{p(\mathbf{a}_t | \mathbf{s}_{[0:t]}, \mathbf{a}_{[0:t-1]}, \boldsymbol{\mu}_k, \boldsymbol{\Lambda}_k, \boldsymbol{\theta}_k)}$.

Here, after each step of policy update, $p(\mathbf{a}_t | \mathbf{s}_{[0:t]}, \mathbf{a}_{[0:t-1]}, \boldsymbol{\mu}, \boldsymbol{\Lambda}, \boldsymbol{\theta})$ from the new policy should be evaluated on the most recent trajectory $\tau_k$ for both next update and approximated KL divergence. However, this quantity can not be calculated directly, but only through sampling $\mathbf{w}_t$ and then integrating $\mathbf{w}_t$ out. Since $\mathbf{w}_t$ is integrated out in the end, it does not matter what specific $\mathbf{w}_t$ is sampled. So one could sample a new set of $\mathbf{w}_t$, or use a fixed $\mathbf{w}$ along the recent trajectory $\tau_k$. The second way is often faster because sampling is avoided. The pseudo-code of single-worker Coherent-PPO is shown in Algorithm 2.

## C.3 COHERENT SOFT ACTOR-CRITIC (COHERENT-SAC)

For Coherent-SAC, only two changes are needed. Firstly, we sample the last layer parameters of the policy network $\mathbf{w}_t$ in each step $t$ for exploration. Secondly, we improve the marginal policy instead of the actual policy performing exploration after each epoch. The pseudo-code of single-worker Coherent-SAC is shown in Algorithm 3.

# D ADDITIONAL RESULTS

In this appendix, we provide additional results for both comparative evaluation and ablation studies.

## D.1 COMPARATIVE EVALUATION

The results of comparative evaluation for A2C (Mnih et al., 2016) and PPO (Schulman et al., 2017) on Reacher-v2, InvertedDoublePendulum-v2 and Hopper-v2 are shown in Figure 4. For SAC (Haarnoja et al., 2018), since it is a state-of-the-art deep RL algorithm, we only test it with three of our six OpenAI MuJoCo continuous control tasks with highest state and action dimensions, as shown in Figure 1.

## D.2 ABLATION STUDIES

The full results of all three ablation studies on all six OpenAI MuJoCo continuous control tasks are shown in Figure 5, Figure 6, Figure 7 and Figure 8.

---

**Algorithm 2:** Coherent-PPO

---

**Input:** initial policy parameters $\boldsymbol{\mu}_0, \boldsymbol{\Lambda}_0, \boldsymbol{\theta}_0$, initial value function parameters $\boldsymbol{\phi}_0$.

1 **for** *k=0,1,2,...,K* **do**

2     Create a buffer $\mathcal{D}_k$ for collecting a trajectory $\tau_k$ with $T$ steps.

3     **for** *t=0,...,T* **do**

4        **if** *t=0* **then**

5           Sample last layer parameters of policy network $\mathbf{w}_t \sim p_0(\mathbf{w}_t; \boldsymbol{\mu}_k, \boldsymbol{\Lambda}_k)$ and store $\mathbf{w}_t$.

6        **else**

7           Sample last layer parameters of policy network $\mathbf{w}_t \sim p(\mathbf{w}_t|\mathbf{w}_{t-1}; \boldsymbol{\mu}_k, \boldsymbol{\Lambda}_k)$ and store $\mathbf{w}_t$.

8        Observe state $\mathbf{s}_t$ and select action $\mathbf{a}_t \sim \pi_{\mathbf{w}_t, \boldsymbol{\theta}_k}(\mathbf{a}_t|\mathbf{s}_t)$.

9        Execute $\mathbf{a}_t$ in the environment.

10       Observe next state $\mathbf{s}_{t+1}$, reward $r_t$, done signal $d$, and store $(\mathbf{s}_t, \mathbf{a}_t, r_t, \mathbf{s}_{t+1}, d)$ in buffer $\mathcal{D}_k$.

11       If $\mathbf{s}_{t+1}$ is terminal, reset environment state.

12       Infer forward message $\alpha(\mathbf{w}_t)$ using previous state $\mathbf{s}_{t-1}$, previous action $\mathbf{a}_{t-1}$ as well as mean $\mathbf{v}_{t-1}$ and covariance $\mathbf{L}_{t-1}^{-1}$ of previous forward message $\alpha(\mathbf{w}_{t-1})$.

13       Store mean $\mathbf{v}_t$ and covariance $\mathbf{L}_t^{-1}$ of current forward message $\alpha(\mathbf{w}_t)$.

14       Compute marginal action probability $p(\mathbf{a}_t|\mathbf{s}_{[0:t]}, \mathbf{a}_{[0:t-1]}, \boldsymbol{\mu}_k, \boldsymbol{\Lambda}_k, \boldsymbol{\theta}_k)$.

15     Compute rewards-to-go $R_t$ and any kind of advantage estimates $\hat{A}_t$ based on current value function $V_{\boldsymbol{\phi}_k}$ for all steps $t$.

16     Learn policy by maximizing the PPO-Clip objective:

$$L_{\boldsymbol{\mu}_k, \boldsymbol{\Lambda}_k, \boldsymbol{\theta}_k}^{CLIP}(\boldsymbol{\mu}, \boldsymbol{\Lambda}, \boldsymbol{\theta}) = \sum_{t=0}^{T-1} \left[ \min\left(r_t(\boldsymbol{\mu}, \boldsymbol{\Lambda}, \boldsymbol{\theta}), \text{clip}\left(r_t(\boldsymbol{\mu}, \boldsymbol{\Lambda}, \boldsymbol{\theta}), 1-\epsilon, 1+\epsilon\right)\right) \hat{A}_t \right],$$

and update the policy by performing multiple gradient steps until the constraint of approximated KL divergence being satisfied:

$$\boldsymbol{\mu}_{k+1} \leftarrow \boldsymbol{\mu}_k + \alpha_{\boldsymbol{\mu}} \hat{\nabla}_{\boldsymbol{\mu}} L_{\boldsymbol{\mu}_k, \boldsymbol{\Lambda}_k, \boldsymbol{\theta}_k}^{CLIP}(\boldsymbol{\mu}, \boldsymbol{\Lambda}, \boldsymbol{\theta})$$

$$\boldsymbol{\Lambda}_{k+1} \leftarrow \boldsymbol{\Lambda}_k + \alpha_{\boldsymbol{\Lambda}} \hat{\nabla}_{\boldsymbol{\Lambda}} L_{\boldsymbol{\mu}_k, \boldsymbol{\Lambda}_k, \boldsymbol{\theta}_k}^{CLIP}(\boldsymbol{\mu}, \boldsymbol{\Lambda}, \boldsymbol{\theta})$$

$$\boldsymbol{\theta}_{k+1} \leftarrow \boldsymbol{\theta}_k + \alpha_{\boldsymbol{\theta}} \hat{\nabla}_{\boldsymbol{\theta}} L_{\boldsymbol{\mu}_k, \boldsymbol{\Lambda}_k, \boldsymbol{\theta}_k}^{CLIP}(\boldsymbol{\mu}, \boldsymbol{\Lambda}, \boldsymbol{\theta}).$$

17     Learn value function by minimizing the regression mean-squared error:

$$L(\boldsymbol{\phi}) = \frac{1}{T} \sum_{t=0}^{T} \left(V_{\boldsymbol{\phi}_k}(\mathbf{s}_t) - R_t\right)^2,$$

and update the value function by performing a gradient step:

$$\boldsymbol{\phi}_{k+1} \leftarrow \boldsymbol{\phi}_k + \alpha_{\boldsymbol{\phi}} \hat{\nabla}_{\boldsymbol{\phi}} L(\boldsymbol{\phi}).$$

---

**Algorithm 3:** Coherent-SAC

---

**Input:** initial policy parameters $\boldsymbol{\mu}, \boldsymbol{\Lambda}, \boldsymbol{\theta}$, initial $Q$-function parameters $\phi_1, \phi_2$, empty replay buffer $\mathcal{D}$.

1 Set target parameters equal to main parameters $\phi_{\text{targ},1} \leftarrow \phi_1, \phi_{\text{targ},2} \leftarrow \phi_2$.

2 **for** *each step* **do**

3    **if** *just updated* **then**

4       Sample last layer parameters of policy network $\mathbf{w} \sim p_0(\mathbf{w}; \boldsymbol{\mu}, \boldsymbol{\Lambda})$ and store $\mathbf{w}$ as $\mathbf{w}_{\text{prev}}$.

5    **else**

6       Sample last layer parameters of policy network $\mathbf{w} \sim p(\mathbf{w}|\mathbf{w}_{\text{prev}}; \boldsymbol{\mu}, \boldsymbol{\Lambda})$ and store $\mathbf{w}$ as $\mathbf{w}_{\text{prev}}$.

7    Observe state $\mathbf{s}$ and select action $\mathbf{a} \sim \pi_{\mathbf{w}, \boldsymbol{\theta}}(\mathbf{a}|\mathbf{s})$.

8    Execute $\mathbf{a}$ in the environment.

9    Observe next state $\mathbf{s}'$, reward $r$, done signal $d$, and store $(\mathbf{s}, \mathbf{a}, r, \mathbf{s}', d)$ in replay buffer $\mathcal{D}$.

10    If $\mathbf{s}'$ is terminal, reset environment state.

11    **if** *it's time to update* **then**

12       **for** *j in range(number of updates)* **do**

13          Randomly sample a batch of transitions $\mathcal{B} = \{(\mathbf{s}, \mathbf{a}, r, \mathbf{s}', d)\}$.

14          Compute targets for $Q$-functions:

$$y(r, \mathbf{s}', d) = r + \gamma(1-d) \left( \min_{i=1,2} Q_{\phi_{\text{targ},i}}(\mathbf{s}', \tilde{\mathbf{a}}') - \alpha \log \pi_{\mathbf{w}, \boldsymbol{\theta}}(\tilde{\mathbf{a}}'|\mathbf{s}') \right),$$

         where $\tilde{\mathbf{a}}' \sim \pi_{\mathbf{w}, \boldsymbol{\theta}}(\tilde{\mathbf{a}}'|\mathbf{s}')$.

15          Update $Q$-functions by one step of gradient descent using:

$$\nabla_{\phi_i} \frac{1}{|\mathcal{B}|} \sum_{(\mathbf{s}, \mathbf{a}, r, \mathbf{s}', d) \in \mathcal{B}} (Q_{\phi_i}(\mathbf{s}, \mathbf{a}) - y(r, \mathbf{s}', d))^2, \quad \text{for } i = 1, 2.$$

16          Update policy by one step of gradient ascent using:

$$\nabla_{\boldsymbol{\mu}, \boldsymbol{\Lambda}, \boldsymbol{\theta}} \frac{1}{|\mathcal{B}|} \sum_{\mathbf{s} \in \mathcal{B}} \left[ \min_{i=1,2} Q_{\phi_i}(\mathbf{s}, \tilde{\mathbf{a}}) - \alpha \log p(\tilde{\mathbf{a}}|\mathbf{s}, \boldsymbol{\mu}, \boldsymbol{\Lambda}, \boldsymbol{\theta}) \right],$$

         where $\tilde{\mathbf{a}}$ is a sample from $p(\tilde{\mathbf{a}}|\mathbf{s}, \boldsymbol{\mu}, \boldsymbol{\Lambda}, \boldsymbol{\theta})$ which is differentiable w.r.t $\boldsymbol{\mu}, \boldsymbol{\Lambda}, \boldsymbol{\theta}$ via the reparameterization trick.

17          Update target networks with:

$$\phi_{\text{targ},i} \leftarrow \rho \phi_{\text{targ},i} + (1-\rho)\phi_i \quad \text{for } i = 1, 2.$$

18    **else**

19       Continue.

---

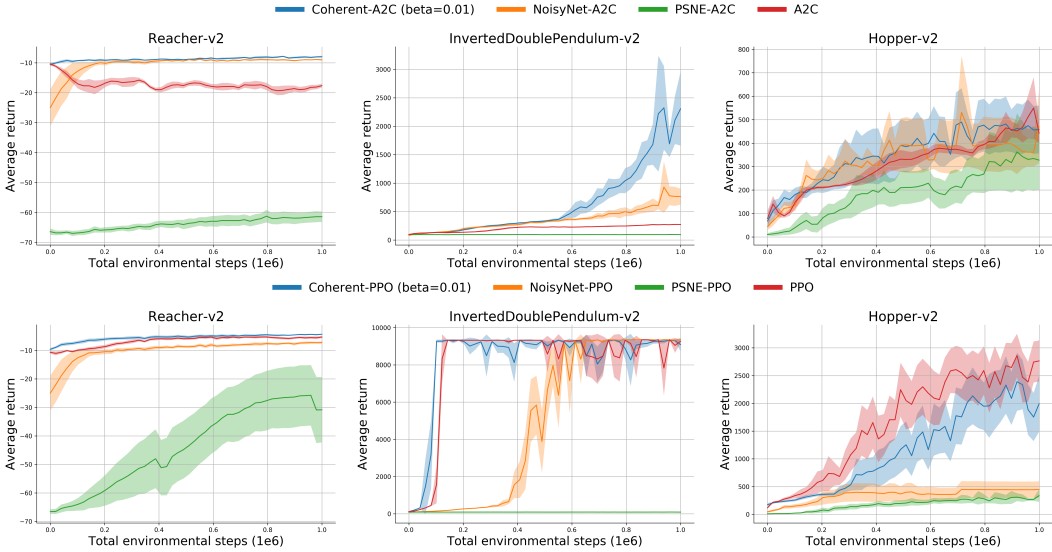

Figure 4: Learning curves for deep RL algorithms with different exploration strategies, where the top and bottom row corresponds to results of A2C and PPO respectively. The solid curves correspond to the mean, and the shaped region represents half a standard deviation of the average return over 10 random seeds.

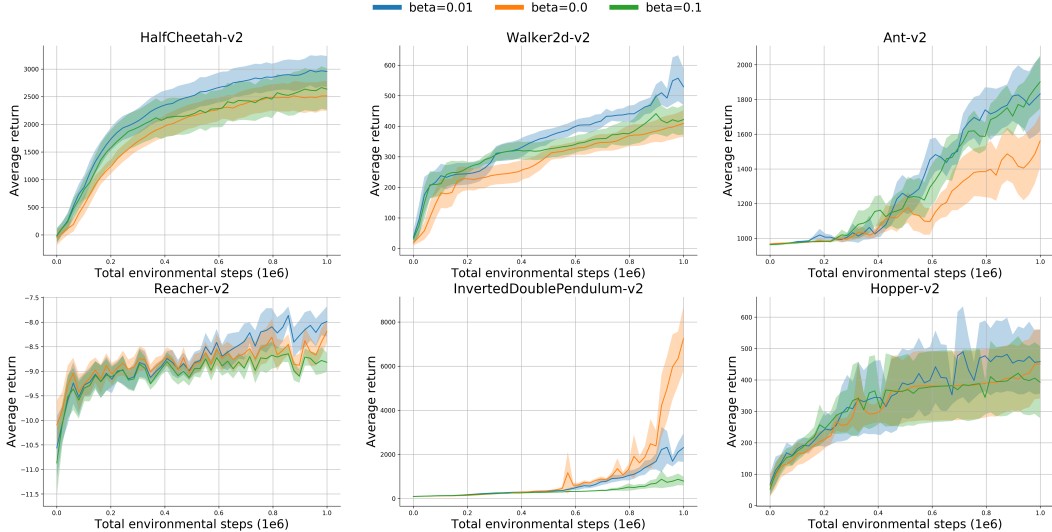

Figure 5: Learning curves for Coherent-A2C on OpenAI MuJoCo continuous control tasks. The solid curves correspond to the mean, and the shaped region represents half a standard deviation of the average return over 10 random seeds.

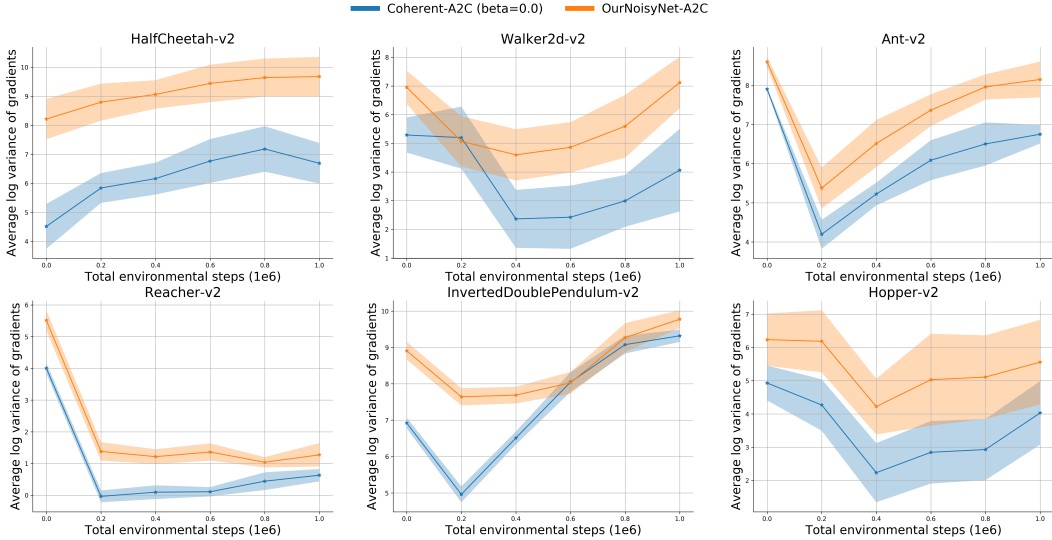

Figure 6: Log variance of gradient estimates for Coherent-A2C and OurNoisyNet-A2C on OpenAI MuJoCo continuous control tasks. The solid curves correspond to the mean, and the shaped region represents half a standard deviation of the average log variance over 10 random seeds.

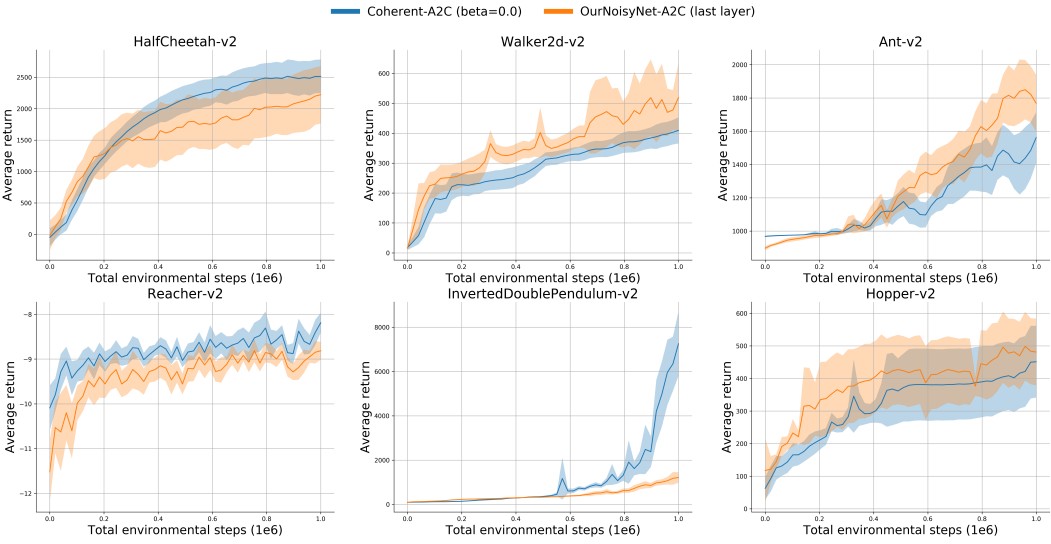

Figure 7: Learning curves for Coherent-A2C and OurNoisyNet-A2C on OpenAI MuJoCo continuous control tasks. The solid curves correspond to the mean, and the shaped region represents half a standard deviation of the average return over 10 random seeds.

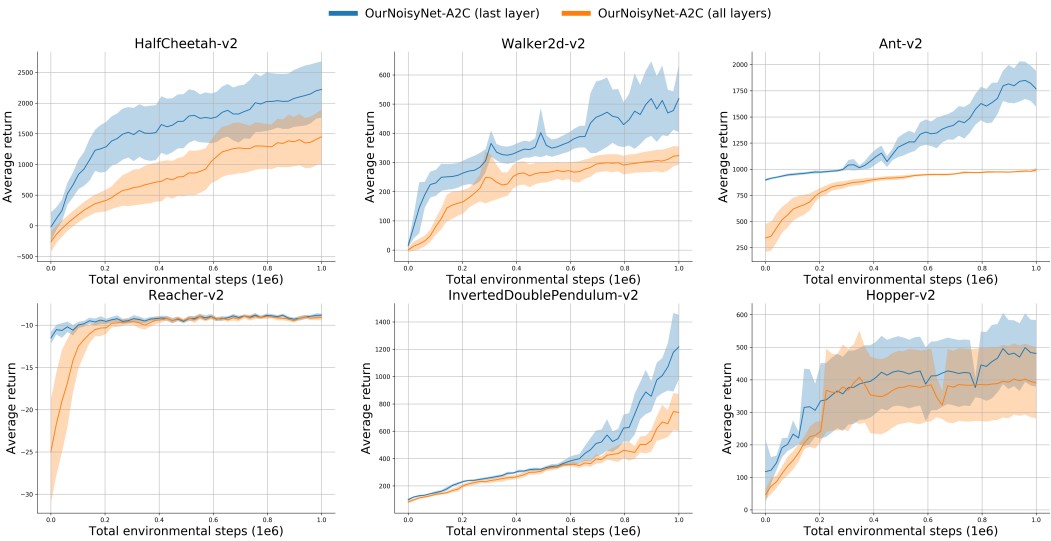

Figure 8: Learning curves for OurNoisyNet-A2C on OpenAI MuJoCo continuous control tasks. The solid curves correspond to the mean, and the shaped region represents half a standard deviation of the average return over 10 random seeds.

