# OpenReview forum: "Deep Coherent Exploration For Continuous Control"
_ICLR.cc/2021/Conference — Reject_

### Official Review · AnonReviewer4 · 2020-10-27
**Deep Coherent Exploration For Continuous Control**

**Rating:** 4
**Confidence:** 4

**Review:**

Summary:

This paper proposes Deep Coherent Exploration that unifies step-based exploration and trajectory-based exploration on continuous control. There exists a prior work that bridges a gap between the two exploration methods for linear policies, and this paper generalizes the prior work for various deep RL methods: on-policy (A2C, PPO) and off-policy (SAC). Finally, Deep Coherent Exploration enhances the performance of baseline algorithms and has better performance than prior works (NoisyNet, PNSE) on Mujoco tasks.

Pros:

+ For combining the proposed method with on-policy learning, this paper derives the log-likelihood of whole trajectory recursively.
+ For on-policy methods (A2C, PPO), the proposed method has large performance gain on Mujoco tasks.

Cons:

- The idea of this paper directly follows GE [van Hoof et al., 2017] and is not much different from GE.
- For SAC, the proposed method is not much effective and it even degrades the performance of the HalfCheetah task.
- The paper focuses on exploration, but the experiments only focus on the return performance of simple Mujoco tasks.
- In order to show the superiority of the proposed method, additional experiments on pure exploration or sparse rewarded tasks are needed.

Minor concerns:

* In background, there is no explanation about step-based and trajectory-based exploration.
* For the off-policy case, there is insufficient explanation for why they use single sigma and the connection point of the proposed method and eq (5).

---

> ### Author Response · Authors · 2020-11-16
> **Response to R4**
>
> We appreciate your time and comments. We are happy that you found our method’s scalability an important contribution and our approach to derive non-Markov policy gradients valuable. We hope to address your critiques below.
>
> 1. Relationship to  GE (van Hoof et al., 2017): We acknowledge that GE (van Hoof et al., 2017) is an important characteristic of our method, but we believe that our recursive and analytical update for non-Markov policy gradient and the architecture of injecting noise in the last layer are valuable additions. As these two characteristics make GE more scalable for DRL algorithms. We also perform detailed ablations that more precisely explain how the various aspects of our method contribute to the performance difference compared to ‘NoisyNets’ and ‘PSNE’.
> 2. SAC-related concerns: We propose a new formulation in the general response and we will provide the experimental results soon. The degradation of our method (only in HalfCheetah) could be due to the “heuristic” approach in SAC, however, it does show slight improvements over the other two Mujoco tasks (Walker and Ant).
> 3. Limitation of experiments to MuJoCo tasks: As in our reply to R2, we want to clarify that our proposal concerns an undirected exploration method. Directed exploration methods target ‘hard exploration’ problems like Montezuma’s revenge, where undirected methods do not have a chance. Still, we believe that improving the exploration behavior of undirected methods is a relevant challenge: undirected methods are commonly used due to their relatively low complexity and easier implementation.  Also, even though all experiments are conducted using the MuJoCo physics simulator, the different tasks differ significantly in nature.
> 4. Explanation about step-based and trajectory-based exploration: We agree we could make them clearer and we will make the change in the updated version of the paper.
>
> Thank you for your time and feedback and we look forward to further discussion.

---

### Official Review · AnonReviewer3 · 2020-10-27

**Rating:** 7
**Confidence:** 2

**Review:**

This paper presents a method to combine step-based exploration with trajectory-based exploration (in the form of action-space noise and parameters-space noise) in continuous MDPs, which is scalable to deep RL methods.

The paper is overall well-written and easy to follow. The Introduction and Related-work sections are good and clear.
Section 3 could benefit from some proof-reading. In particular, Section 3.2 is quite dense. I think it would be unhelpful for the reader not already familiar with the discussed algorithms, and on the other hand redundant for those familiar with them. I would consider moving it to the appendix, and instead provide a more high-level description of policy-gradient methods, without getting into the specific details of PPO vs SAC vs A2C. Also consider that this section uses terms which are not explicitly defined (Q-function and Advantage function) again, making it less approachable or clear for readers less familiar with RL.
One other minor (and technical) issue is that the font used in the figures (legend, axis titles, etc) is very small, barely readable even in 150%.

While the underlying theoretical ideas are not novel (as the authors mention, the basic approach here is following Hoof et al. 2017), there is an important contribution in the scalability of the method, as well as in its evaluation on "standard" benchmark for continuous RL against some other strong baselines. Another important advantage of the approach is that while the policy is non-markov (due to the "global" trajectory-based exploration or coherence), the policy gradients can still be estimated in a more-or-less standard, step-based, way, thanks to analytical integration of the "latent" variables (basically the parameters of the last layer), hereby overcoming the challenge of high variance in PG estimate for non-markov policies.

---

> ### Author Response · Authors · 2020-11-16
> **Response to R3**
>
> We are appreciated for your time and your positive comments. We are happy that you found our method’s scalability an important contribution and our approach to derive non-Markov policy gradients valuable. We hope to respond to your suggestions as follows.
>
> 1. Moving the section introducing DRL algorithms to the appendix and instead provide an introduction of policy gradients: We agree with your suggestion and we will work on improving the presentation of our contents.
> 2. Enlarging the font size in the figures: We agree and we will increase the font size in the updated version of the paper.
>
> Thank you for your time and feedback and we look forward to further discussion.

---

### Official Review · AnonReviewer2 · 2020-10-28
**Complex approach with little improvement**

**Rating:** 4
**Confidence:** 3

**Review:**

Summary: This paper focuses on undirected exploration strategies in reinforcement learning. Following the prior work, this paper proposes an exploration method unifying the step-based and trajectory-based exploration. The authors propose to perturb only the last(linear) layer of the policy for exploration, instead of perturbing all layers of the policy network. Also, the authors use analytical and recurrent integration for policy updates. Experiments show that the proposed exploration strategy mostly helps A2C, PPO and SAC in three Mujoco environments.

Clarity:
This paper is generally written clearly. Some details need more clarification as pointed out in 'Cons'.

Originality:
As far as I know, the proposed technique is novel in the literature of undirected exploration. But for the three bullet points in section 1, the first point of "Generalizing Step-based and Trajectory-based Exploration" should not be one of the main contributions of this paper, because this paper follows the formulation of policy in van Hoof et al. (2017) and the latter proposed the generalized exploration connecting step-based and trajectory-based exploration. The work can be viewed as an extension of van Hoof et al. (2017)  with a deep policy network.

Significance:
The proposed method is mathematically solid, but the main concern lies in empirical performance. Nowadays SAC is the state-of-the-art and generally used method for continuous control tasks and it is more advanced than A2C and PPO. But the proposed method does not obviously improve the performance of SAC while inducing much more complexity in policy learning. Therefore the significance of the proposed approach in practice might be limited.

Pros:
*The authors provide detailed mathematical derivation (in the main text and the appendix) to support the proposed method.
*The proposed method significantly outperforms the baselines when investigating the on-policy methods A2C and PPO.
*The authors provide ablative studies about hyper-parameter values and components of the proposed method with A2C.

Cons:
*In section 4.2, "we maintain and adapt a single magnitude σ for the parameter noise". What's the motivation of this setting different from the formulation in section 4.1?
*In section 5, why the advantage of the proposed method is poor with SAC? What's the value of hyper-parameters α and δ? Is the proposed method sensitive to these hyper-parameter choices?
*In section 5, apart from the comparison of the performance of the learned policy, the comparison of the complexity (which might be measured by wall time to learn the policy?) of different exploration strategies can also be interesting.
*In the first two rows of Figure 1, why the baseline methods NoisyNet-A2C(PPO) and PSNE-A2C(PPO) even significantly underperform the vanilla A2C(PPO)? The intuition is that introducing exploration strategies will mostly help the agent learns more quickly. Is it possible that the baselines are not tuned well?
*The experiments on a single domain (Mujoco) seems not convincing enough. It will be better if there are experiments on other more complicated domains.

---

> ### Author Response · Authors · 2020-11-16
> **Response to R2**
>
> Thank you for your time and comments. We are happy that you found our approach novel and our method mathematically solid. Please consider the following rebuttals to your concerns.
>
> 1. SAC with Deep Coherent Exploration: We propose a new formulation in the general response and we will provide the experimental results soon.
> 2. Claimed contribution: We agree that the first bullet point in the introduction is not a contribution of the paper. This is one of three key properties that, as we stated in the paper, $\textbf{together}$ define our contribution. We will further clarify that in the paper.
> 3. Coherent-SAC and complexity in policy learning: In our current Coherent-SAC formulation, relatively little complexity is added, as the only complexity added is sampling the parameters and adapting the variance. The new Coherent-SAC formulation we are currently working on will add some complexity, but less than the on-policy method as we only need the per time-step marginal, rather than conditioning on the history so far. This key difference is due to the way SAC optimizes its policy. Since SAC tries to make its policy close to the softmax of the Q-function, the probability of action given the history is naturally not involved. Thus the recurrent inference is then not necessary.
> 4. Significance of the proposed approach in practice: PPO is also a popular and widely-used DRL algorithm and we believe the increased performance of coherent-PPO compared to PPO can also be valuable in practice.
> 5. Motivation of SAC setting different from the formulation in section 4.1: As stated in the general response, we attempted to stay close to (Plappert et al., 2018) but are currently working on a more integrated version of coherent-SAC.
> 6. Performance of coherent-SAC: We expect a more integrated version of Coherent-SAC to yield better results. We expect to share these later this week.
> 7. What's the value of hyper-parameters α and δ: We set α=1.01 and δ=σ=0.1 as in Parameter Space Noise for Exploration (Plappert et al., 2018).
> 8. Comparison of the complexity of different exploration strategies: Each update has a complexity linear in the number of time steps, and our method requires an inverse of a matrix that grows with the number of weights d_last in the last layer, so scaling cubically in d_last with a naive algorithm.
> 9. Poor performance and tuning of NoisyNet-A2C(PPO), PSNE-A2C(PPO): Firstly, NoisyNet-A2C and PSNE-A2C perform similarly compared to A2C, except that PSNE-A2C performs poorly in Ant-v2. It’s not clear why this happens but similar performance is also reported in Plappert et al., 2018, showing that PSNE is very sensitive to hyperparameters. For NoisyNet, its performance for continuous control tasks has not been reported before. For PPO, we believe there are two reasons for the uncompetitive performance of NoisyNet and PSNE. The first reason is that both NoisyNet and PSNE have more parameters than our method as they perturb the policy over all layers. As a result, the updated policy can diverge faster from the old policy and hence meet the KL constraint in PPO much earlier, resulting in much fewer updates than the vanilla PPO. We observed this reduction in update steps in PPO in our experiments. Moreover, since PSNE adapts the scale “hard” and heuristically, the difference could be even severe. The second reason concerns the variance of the gradient estimates. As shown in our paper, NoisyNet’s gradient estimates have much higher variance, thus leading to more oscillating updates of the parameters and hence could result in more different policy in terms of KL divergence. Lastly, we did not explicitly tune the hyperparameters of all methods. But we will consider doing that.
> 10. The experiments on a single domain (Mujoco) seems not convincing enough:
> First, we consider the Mujoco environments to be quite different from each other. The fact that they use the same underlying physics simulation does not change that in our opinion. Second, we want to clarify that our proposal concerns an undirected exploration method. Directed exploration methods target ‘hard exploration’ problems like Montezuma’s revenge, where undirected methods do not have a chance. Still, we believe that improving the exploration behavior of undirected methods is a relevant challenge: undirected methods are commonly used due to their relatively low complexity and easier implementation.
>
> Again, we want to thank you for your time and comments and we look forward to further discussion.

---

### Official Review · AnonReviewer1 · 2020-10-28
**A promising exploration method that would be of interest to many in the community.**

**Rating:** 7
**Confidence:** 3

**Review:**

I would like to thank the authors of "Deep Coherent Exploration For Continuous Control" for their valuable and interesting submission.

Summary of the paper
--

The basis of this work is van Hoof et al., 2017; there, “Generalized Exploration” views policy parameters as being drawn from a per-trajectory Markov chain. Experience is collected with a different set of parameters at each timestep, corresponding to steps along the chain.
The authors of this work introduce “Deep Coherent Exploration”, which scales to deep reinforcement learning methods.
The main contributions are:
1. Simplifying the setting by modeling just the parameters in the last layer.
2. A recursive, analytic expression for marginalizing over the last-layer parameters, useful for obtaining low-variance gradients with on-policy methods.
3. Detailed recipes for incorporating the method into on-policy methods (A2C and PPO) as well as an off-policy method (SAC).

Assessment
--
This work explores an important problem in RL and proposes a promising method that would be of interest to many in the community, and I think it would be a valuable contribution to ICLR.

-- The positives --

The paper is well written, and does a great job of introducing the reader to the relevant concepts and situating itself in the literature.
The empirical results for the on-policy methods are really strong and clearly demonstrate the value of this approach. Additionally, the ablation experiments were very insightful.
The detailed appendix makes me confident that readers would be able to easily reproduce the method.

-- The concerns --

The story for off-policy methods seems almost unrelated: the generative model is much more restrictive (isotropic noise), the optimization method is based on a heuristic (that subsequent policy parameters should be separated by a fixed distance in the action distribution), detailed balance isn't maintained within the Markov chain, and the experimental results aren’t as strong as those of the on-policy settings.

Suggestions
--
It might make more sense to reframe this as an on-policy method and explicitly address the off-policy case as a limitation. Would the authors consider this alternative?

I tentatively score this paper as accept, and looking forward to the rebuttal to calibrate with the other reviewers.

---

> ### Author Response · Authors · 2020-11-16
> **Response to R1**
>
> Thank you for taking the time to read our paper and provide us with feedback. We are glad that you found our approach valuable and our experiments insightful. We hope to address your concerns below.
>
> 1. SAC with Deep Coherent Exploration: We propose a new formulation in the general response and we will provide the experimental results soon.
> 2. Detailed balance within the Markov chain: We might not have explained this clearly enough in the paper. The generative process of exploration noise for a particular set of parameters is the same as for on-policy method, and thus preserves detailed balance. However the base distribution (parameterized by mu, Lambda), is set (learned or provided by a heuristic), eq. (5) ensures all marginals are equal to this base distribution.
>
> Again, we want to thank you for your time and comments and we look forward to further discussion.

---

### Author Response · Authors · 2020-11-16
**General response for SAC**

We thank all reviewers for your detailed and valuable comments. Here we would like to respond to one common concern about SAC, including how we integrate SAC with our method and the empirical results.
1. Integrating SAC with Deep Coherent Exploration: Except for the different ways of policy update in on-policy methods and off-policy methods, we chose this specific heuristic approach to combine our method with SAC in the beginning because this exact same way has been applied and was proven effective in Parameter Space Noise for Exploration (Plappert et al., 2018). Note that this is only for the policy update, for exploration, it is still unchanged and the same with on-policy methods.
2. However, we agree with the reviewers that our integration of SAC is more heuristic than our integration of on-policy methods like A2C and PPO. Considering that, we propose a new mathematical formulation of Coherent-SAC. To do that, we closely follow the idea of policy update in SAC: we now optimize our policy by minimizing the KL divergence of the marginalized policy (the policy that could have been sampled on average) at step t to the softmax distribution of Q-function. This integration is natural to both Deep Coherent Exploration and SAC. Moreover,  this integration with SAC is more efficient than with on-policy methods: the Gaussian marginalization does not require matrix inversion. This integration will also allow the variance of the distribution over parameters to be learned rather than heuristically set, which we expect will improve results.
3. Empirical results of Coherent-SAC: We are currently working on the experiments and expect to add and show the results in the paper in a short time. If this method performs as we expect, we will update the corresponding text in our paper.

---

> ### Author Response · Authors · 2020-11-25
> **Manuscript updated**
>
> We would like to note that we have now updated the manuscript. In particular, we describe a way to use coherent exploration with SAC that is more consistent with the approach for the on-policy methods, and yields slightly better results. The approach also allows us to update the precision $\Lambda$ of the search distribution rather than set it heuristically. The major changes are in Sec 4.2, the bottom row of Fig 1, and the associated discussion in 5.1. We have not yet been able to fully address the comments regarding presentation, which we'll finish for a camera-ready version.

---

### Decision · Program_Chairs · 2021-01-07
**Final Decision**

**Decision:**

Reject

**Comment:**

Unfortunately some of the reviewers' reactions to the author feedback won't be visible to the authors.
The reviewers highly appreciated the replies and revision of the paper

Pros:
- The paper renders Generalized Exploration tractable for deep RL.
- The idea is applicable to many DRL methods and is potentially very valuable to deal with the headaches associated to DRL.

Cons:
- R2 and R4 are still concerned about whether 'smart' exploration will always be advantageous, and whether the added complexity is a good trade-off for the (potentially) better performance. A comparison to 'pure' exploration would still be insightful.
- the new 'SAC with Deep Coherent Exploration' only partially addresses the concerns of R2 and R4, especially in terms of performance

While the paper has improved drastically during the reviewing process, there are still a few too many doubts.